# Spatiotemporal Distribution Characteristics of Reference Evapotranspiration in Shandong Province from 1980 to 2019

**Fujie Zhang [1], Zihan Liu [1,2], Lili Zhangzhong [2,3,\*], Jingxin Yu [2,3], Kaili Shi [2,3] and Li Yao [2,3]**

[1] Faculty of Agriculture and Food, Kunming University of Science and Technology, Kunming 650500, China; 20030031@kust.edu.cn (F.Z.); zihanliu1010@163.com (Z.L.)

[2] National Engineering Research Center for Intelligent Equipment in Agriculture, Beijing Academy of Agriculture and Forestry Sciences, Beijing 100097, China; Jingx.Yu@outlook.com (J.Y.); skaili_91@163.com (K.S.); lilili320yao@163.com (L.Y.)

[3] Key Laboratory for Quality Testing of Hardware and Software Products on Agricultural Information, Ministry of Agriculture, Beijing 100125, China

\* Correspondence: lilizhangzhong@163.com

**Abstract:** Reference evapotranspiration ($ET_0$) is an important part of the water cycle and energy cycle during crop growth. Understanding the influencing factors and spatiotemporal variations of $ET_0$ can guide regional water-saving irrigation and regulate agricultural production. Data for daily meteorological observations of temperature, relative humidity, wind speed, and sunshine hours from 38 surface meteorological stations were used to analyze the spatiotemporal variations and trends in Shandong Province from 1980 to 2019. (1) The $ET_0$ from 1980 to 2019 was 1070.5 mm, and there was a significant downward trend in the climate tendency rate of −7.92/10 a. The climate of Shandong Province became warmer and drier. The average annual temperature showed a significant upward trend, while the average annual relative humidity and average annual sunshine hours showed significant downward trends. (2) The annual $ET_0$ ratio in spring, summer, autumn, and winter was 29%, 40%, 21%, and 10%, respectively. (3) A change in Shandong Province's interannual $ET_0$ occurred in 2002, with a decrease of 130.74 mm since then. (4) The $ET_0$ was positively correlated with temperature, wind speed, and sunshine hours and negatively correlated with relative humidity. This study provides a scientific basis for the regulation and control of agricultural production in Shandong Province.

**Keywords:** reference crop evapotranspiration; temporal and spatial distribution; meteorological factors; Shandong Province; Pearson correlation

## 1. Introduction

Crop evapotranspiration refers to the process of leaf water transpiration and soil water evaporation during the whole growth period of crops from sowing to maturity [1]. Reference crop evapotranspiration ($ET_0$), as an important index to characterize potential evapotranspiration and to evaluate the degree of drought, water consumption by vegetation, production potential, and water supply and demand balance [2,3], plays an important role in the water and energy cycles during crop growth [4]. $ET_0$ is an important element to understand global climate change and improve the utilization of agricultural water resources [5].

The amount of $ET_0$ is likely affected to meteorological factors and crop conditions, which are the main influences on $ET_0$ change and their influence will increase with global climate change [6]. The spatiotemporal distribution of $ET_0$ trends and its influencing factors have been a focus of research

within and outside China. From the point of international research, the $ET_0$ in Spain showed a significant downward trend from 1961 to 2011, mainly because of a decrease in relative humidity and a sustained and significant increase in wind speed [7]. Valentin et al. [8] found that an increase in temperature and precipitation in southern Russia and most parts of the United States caused $ET_0$ to increase. Goyal et al. [9] analyzed the response of $ET_0$ to meteorological factors of Rajasthan in India. They concluded that a decrease in wind speed and an increase in air temperature resulted in a decrease of $ET_0$. In Chinese domestic research, Qi et al. [10] showed that the average annual $ET_0$ of Heilongjiang Province in the past 50 years had an decreasing trend, and the spatial trend gradually decreased from west to east, from south to north. Through an analysis of the spatiotemporal variation of $ET_0$ in Jiangsu Province, Wang et al. [11] found that the $ET_0$ mainly showed a decreasing trend, although the $ET_0$ of southern Jiangsu showed an increasing trend. The average annual $ET_0$ has decreased since the 1980s. The study of spatiotemporal variations in $ET_0$ in the hilly region of central Sichuan by Feng et al. [12] found that the spatial distribution of $ET_0$ showed a decreasing trend from northeast to southeast to central areas. The study of Liu et al. [13] showed that, at different time scales, $ET_0$ in Beijing was mainly affected by different meteorological factors: the most influential factors affecting the annual time scale $ET_0$ are relative humidity and net radiation, and the factors that have a great influence on the daily time scale $ET_0$ are wind speed and temperature. Li et al. [14] analyzed the spatiotemporal characteristics of $ET_0$ on the Loess Plateau of China and predicted its future variation trend. These studies show that $ET_0$ changes have regional characteristics because of different climatic conditions, resulting in different spatiotemporal patterns and characteristics [15]. Shandong Province is located on the east coast of China, with an unstable, warm, temperate monsoon climate. As a large agricultural province in China, grain crops covered 11 million hectares in 2018, with a total output of nearly 150 million tons. However, the effective irrigation coefficient was only 0.47, and the per capita water resources were less than 1/6 of the whole country [16]. Drought and waterlogging have adverse effects on agricultural production and social economic development, so relevant research on spatiotemporal patterns of $ET_0$ is needed [17].

In addition, there are a large number of methods analyzing meteorological data. The Mann–Kendall (M–K) test is a trend analysis method that has been widely used in hydrology and meteorology in recent years. Zhao et al. [18] used the M–K test to study the annual $ET_0$ of Ningxia and found that $ET_0$ showed a downward trend and change occurred around 1990. Zhang et al. [19] showed that the standardized precipitation evapotranspiration index (SPEI) on the Loess Plateau decreased in 1990s. The climate tendency rate is often used to reflect the climate change trend. A positive value shows an upward trend, while a negative value indicates a downward trend. Wang et al. [20] analyzed variations in the temporal trends of $ET_0$ and precipitation in Heilongjiang Province using the climate tendency rate. Zuo et al. [21] showed that the spatial distribution of annual $ET_0$ of the Wei River basin decreased from northeast to southwest, and the spatial variation in $ET_0$ was significant. Interpolation methods have significant advantages in analyzing the spatial distribution characteristics and future trends of climate. Radial basis function (RBF) [22], kriging [23], Bayesian kriging regression (BKR) [24], inverse distance weight (IDW), and other interpolation methods have been widely used. Among them, IDW is a spatial distribution method that fully considers the regional relationship between various factors. Because of its simple principle and accurate results, it has been widely used in data interpolation processing. Many scholars have used the inverse distance weight difference (IDW) method to analyze the spatial distribution of $ET_0$, e.g., Han et al. [25] using IDW to analyze the spatial distribution characteristics of the Tarim basin. Based on the inverse distance weight method, Jia et al. [26] evaluated the applicability of the algorithm for $ET_0$ in the Yangtze River Basin.

From the above analysis, it is clear that the response of meteorological factors to $ET_0$ varies greatly on different spatial and temporal scales. Based on it, this study selected daily meteorological data from 38 surface meteorological stations in Shandong Province covering nearly 40 years. The FAO-56 Penman–Monteith method was used to calculate $ET_0$, and the IDW method was utilized to analyze the spatial distribution of $ET_0$ and meteorological factors. This study analyzed the change in the $ET_0$ trend using the M–K test and examined the Pearson correlation coefficient between meteorological factors

and ET$_0$. Analyzing the spatiotemporal distribution characteristics of ET$_0$ change on different time scales will provide a theoretical basis for water resources regulation, irrigation system design, and crop water management in Shandong Province.

## 2. Materials and Methods

### 2.1. Overview of the Study Area

Shandong Province, one of the coastal provinces in East China, is located in the east coast of China, 34°22.9′–38°24.01′ N, 114°47.5′–122°42.3′ E. It is referred to as Lu, and the provincial capital is Jinan. It is bordered from north to south by Hebei, Henan, Anhui, and Jiangsu provinces. The climate is a warm, temperate monsoon climate, the precipitation and hottest temperatures are concentrated in the summer, the spring and autumn are short, and the winter and summer are long. The research area and site distribution are shown in Figure 1.

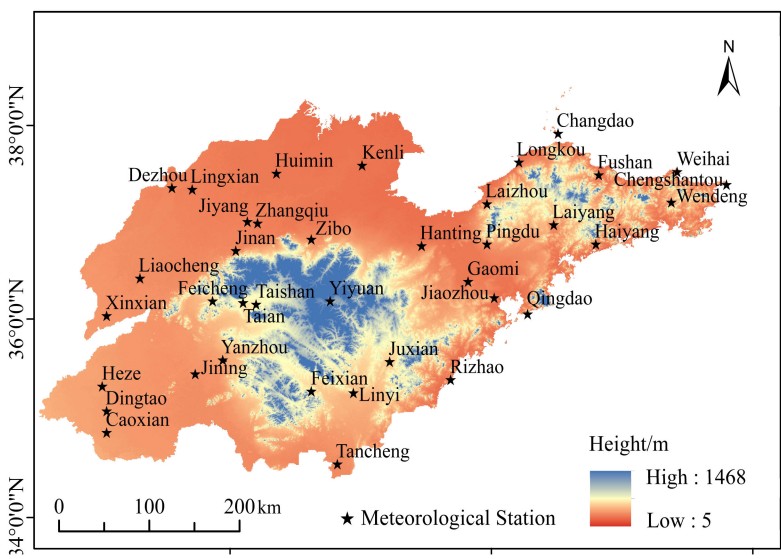

**Figure 1.** Distribution map of meteorological stations in Shandong.

### 2.2. Data Sources and Processing

The meteorological data of this paper were from the China Meteorological Data Network (http://data.cma.cn). Daily meteorological data (maximum temperature, minimum temperature, average temperature, average relative humidity, wind speed, and sunshine hours) from 38 meteorological stations in the research area were selected. The meteorological factors are expressed as $T$max, $T$min, $T$, $RH$, $WS$, and $SSD$, respectively. There were $5.55 \times 10^5$ data points. Where part of the meteorological station daily value data was missing or abnormal, linear interpolation of mean or median value methods were used to supplement the missing data, according to the situation. Geographic information includes the latitude, longitude, and altitude of each site and administrative boundaries of the study area. The range of specific meteorological data is shown in Table 1.

**Table 1.** Data range of meteorological factors.

| Meteorological Factors | $T$ (°C) | $RH$ (%) | $WS$ (m/s) | $SSD$ (h) |
|---|---|---|---|---|
| Data range | −50–50 | 0–100 | 0–20 | 0–20 |
| Data accuracy | 0.1 | 1 | 0.1 | 0.1 |

### 2.3. FAO-56 Penman–Monteith Equation

The Penman–Monteith formula recommended in the 1998 FAO-56 paper was adopted. Based on the principle of energy balance and aerodynamics, the equation has a complete theoretical basis and high calculation accuracy and is widely used across the globe [27]. The Penman–Monteith model is expressed as follows:

$$\text{ET}_0 = \frac{0.408\Delta(R_n - G) + \gamma\frac{900}{T+273}U_2(e_s - e_a)}{\Delta + \gamma(1 + 0.34U_2)} \tag{1}$$

where $\text{ET}_0$ represents the reference crop evapotranspiration, mm/d; net radiation is $R_n$, MJ/(m$^2$·d); $G$ is the soil heat flux density, MJ/(m$^2$·d); mean air temperature is $T$, °C; $e_s$ is the saturated vapor pressure, kPa; actual water vapor pressure is $e_a$, kPa; $\Delta$ is the slope of the saturation vapor pressure function, kPa/°C; the psychometric constant is $\gamma$, kPa/°C; and $U_2$ represents the wind speed at 2 m height, m/s.

### 2.4. Climate Tendency Rate

The climate tendency rate reflects the changing trend of climate elements, which can be calculated using linear regression. The calculation formula of linear regression coefficient (*a*) sample ($Y_i$) and time is as follows:

$$Y_i = at + b \tag{2}$$

where $t$, $b$, and $10 \times a$ are time series, empirical coefficient, and climate tendency rate, respectively.

### 2.5. Mann–Kendall Trend Testing

The M–K method is a non-parametric statistical test. Compared with parameter tests, the advantage of the M–K trend test is that its sample does not need to follow certain distribution requirements nor is it affected by a few outliers, so it is more suitable for type variables and sequential variables. The trend analysis method has been used widely in hydrology and meteorology in recent years [18]. In this paper, we used a confidence level of $\alpha = 0.05$ to test the data and find out the change points through the M–K test and to analyze the trend of meteorological factors and evapotranspiration before, after, and throughout the year.

### 2.6. Pearson Correlation

Pearson correlation coefficients are often used to detect whether independent and dependent variables are correlated. Their values range between −1 and 1. The formula is as follows:

$$P_{x,y} = \frac{\sum(x - \overline{x})(y - \overline{y})}{\sqrt{\sum(x - \overline{x})^2 \cdot \sum(y - \overline{y})^2}} \tag{3}$$

where $\sum(x - \overline{x})(y - \overline{y})$, $\sqrt{\sum(x - \overline{x})^2 \cdot \sum(y - \overline{y})^2}$, and $P_{x,y}$ are the covariance between variable $x$ and variable y, standard deviation between variable $x$ and variable $y$, and the correlation coefficient, respectively. The standard deviation (STDVP), root mean square error (RMSE), and Pearson coefficient of the $\text{ET}_0$ and meteorological factors were calculated, and a Taylor correlation diagram was drawn [28].

### 2.7. Inverse Distance Weighted Interpolation

The IDW method in ArcGIS10.2 (ESRI, Redlands, CA, USA) was used to study the spatial distribution of regional $\text{ET}_0$. IDW is a common and simple spatial interpolation method. It is based on the principle of similarity, that is, the closer the two objects are, the more similar their properties are. The distance between the interpolation point and the sample point is a weighted average, and the weight is given to the sample point [23]. The spatial distribution of $\text{ET}_0$ and meteorological factors in Shandong Province were analyzed using IDW.

## 3. Results

### 3.1. Temporal Variation of Meteorological Factors

The interannual variation of meteorological factors in Shandong Province was uneven, and the values showed noticeable oscillations (Figure 2). *T*, *T*max, and *T*min all had an upward trend during the period 1980−2019, with a range of 2.5−3.5 °C and a climatic tendency rate of 0.37, 0.26, and 0.51 °C, respectively. the average annual *T* was 13.2 °C. Overall, the annual average *RH* was 66.5%, the climate tendency was −0.44%/10 a, and there was no obvious change trend. The overall average *WS* at 2 m showed a downward trend below the average of 2.16 m/s at the beginning of the 21st century, with an average reduction of 0.179 m/s per 10 years. The maximum *SSD* was 7.3 h in 1981, the minimum was 5.9 h in 2003, and the average over the years was 6.6 h, with an average reduction of 0.263 h per 10 years.

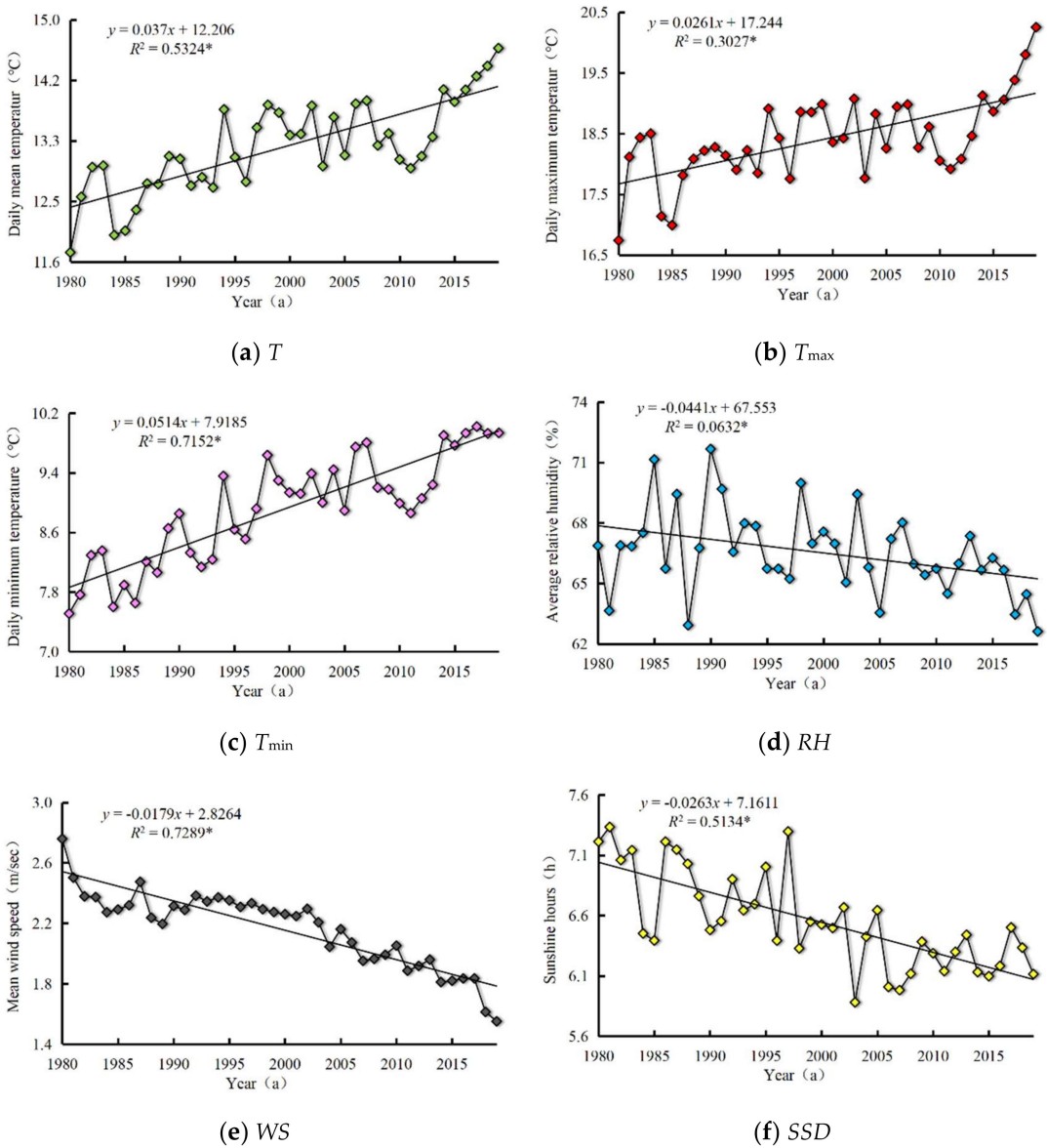

**Figure 2.** Temporal distribution trends of meteorological factors (**a**) daily mean temperature (*T*), (**b**) maximum temperature (*T*max); (**c**) minimum temperature (*T*min); (**d**) average relative humidity (*RH*); (**e**) wind speed at 2 m (*WS*); and (**f**) sunshine hours (*SSD*). Note * indicates a significant level of α = 0.05.

## 3.2. Spatial Distribution of Meteorological Factors

It can be seen from Figure 3 that the distribution of meteorological factors in Shandong Province was spatially variable. The *RH* distribution had latitudinal zonal characteristics, and the *T*, *WS*, and *SSD* differed with longitude. The average *T* range was 6.0–14.9 °C, which is relatively large. The highest-value areas were in southwest Shandong and parts of northern Shandong, with an average of 14.3 °C. Jiaodong, which is close to the Bohai Sea, and the Taishan area, which is at high altitude, had a low average *T* of only 10.7 °C. The *RH* ranged from 56.8% to 74.2%, showing a relatively large difference. The high-value areas were mainly distributed in Chengshantou on the Jiaodong Peninsula and Cao County in southwest Shandong, at 74.2% and 71.3%, respectively. The low-value centers were distributed in Jinan and Zibo in northern Shandong, at 56.8% and 61.3%, respectively. The average *WS* ranged between 1.25 and 5.09 m/s. Taishan Station had a high *WS* because of its high altitude, while the other high-value centers appeared on the Jiaodong Peninsula along the coast, especially Chengshantou, Long Island, and Weihai, with *WS* of 4.4, 4.8, and 3.5 m/s; respectively. The total range of *SSD* was 5.9–7.3 h. The highest-value locations were 7.25, 7.24, and 7.12 h in Longkou, Kenli, and Laizhou, while Liaocheng, Tancheng, and Xinxian had a lower annual *SSD*, with an average of 5.94 h.

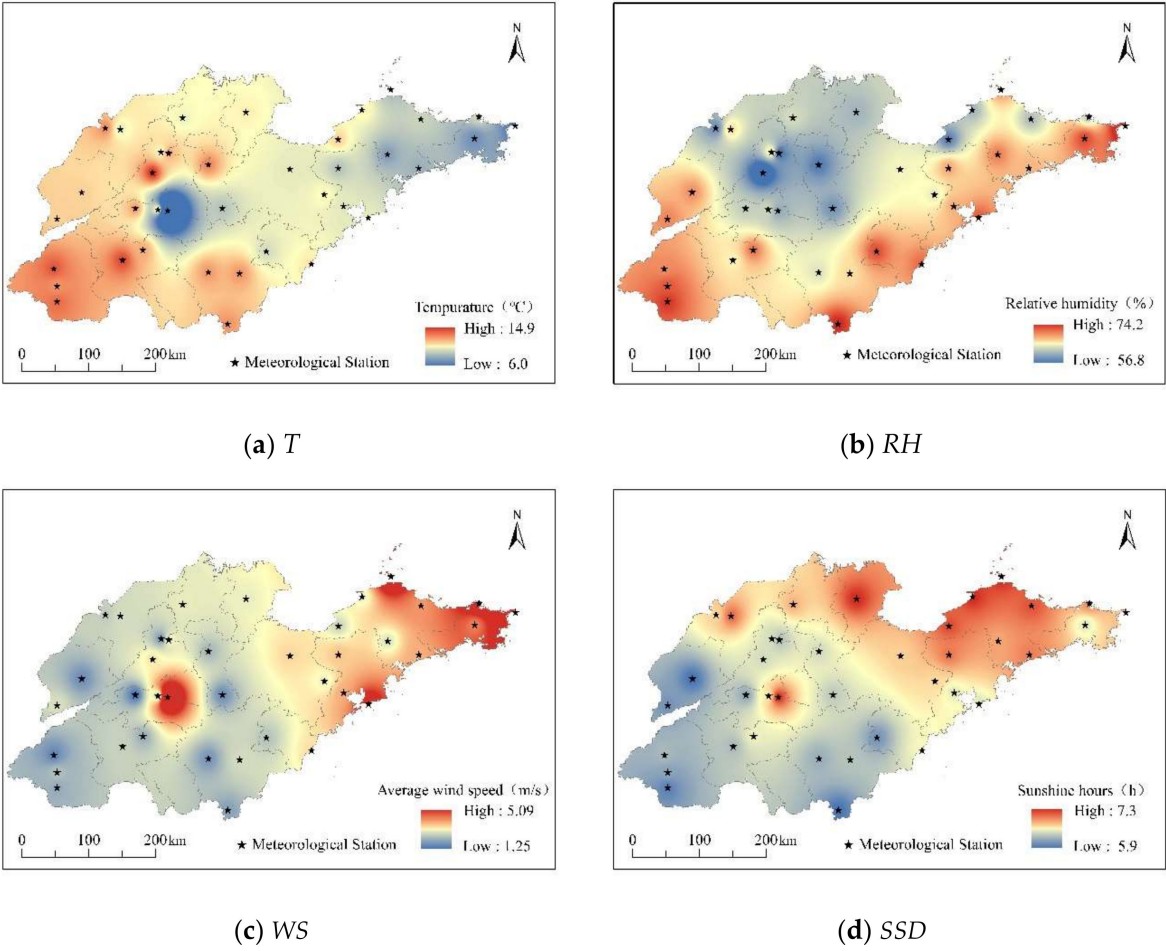

**Figure 3.** Spatial distribution of meteorological factors (**a**) daily mean temperature (*T*), (**b**) average relative humidity (*RH*); (**d**) wind speed at 2 m (*WS*); and (**d**) sunshine hours (*SSD*).

## 3.3. Temporal Variation of $ET_0$

The average annual $ET_0$ of Shandong Province from 1980 to 2019 showed a downward trend, at a rate of −7.92 mm/10 a. The maximum interannual $ET_0$ was in 1981, at 1161.85 mm; the lowest was in 1985, at 992.77 (Figure 4a). There was a fluctuating downward trend up to 2010, with a slow upward

trend after that. With respect to inter-decadal change, as shown in Figure 4b, the $ET_0$ declined from the 1980s to the 2010s, with a decline of 9.38, 5.88, and 13.11 mm per decade. The 1980s and 1990s showed a positive anomaly, which was largest in the 1980s. The 2000s and the 2010s showed a negative anomaly, which was largest in the 2010s. The accumulated anomaly of annual $ET_0$ was variable but generally upward, as shown in Figure 4c, peaking in 2002 and then gradually decreasing until 2016.

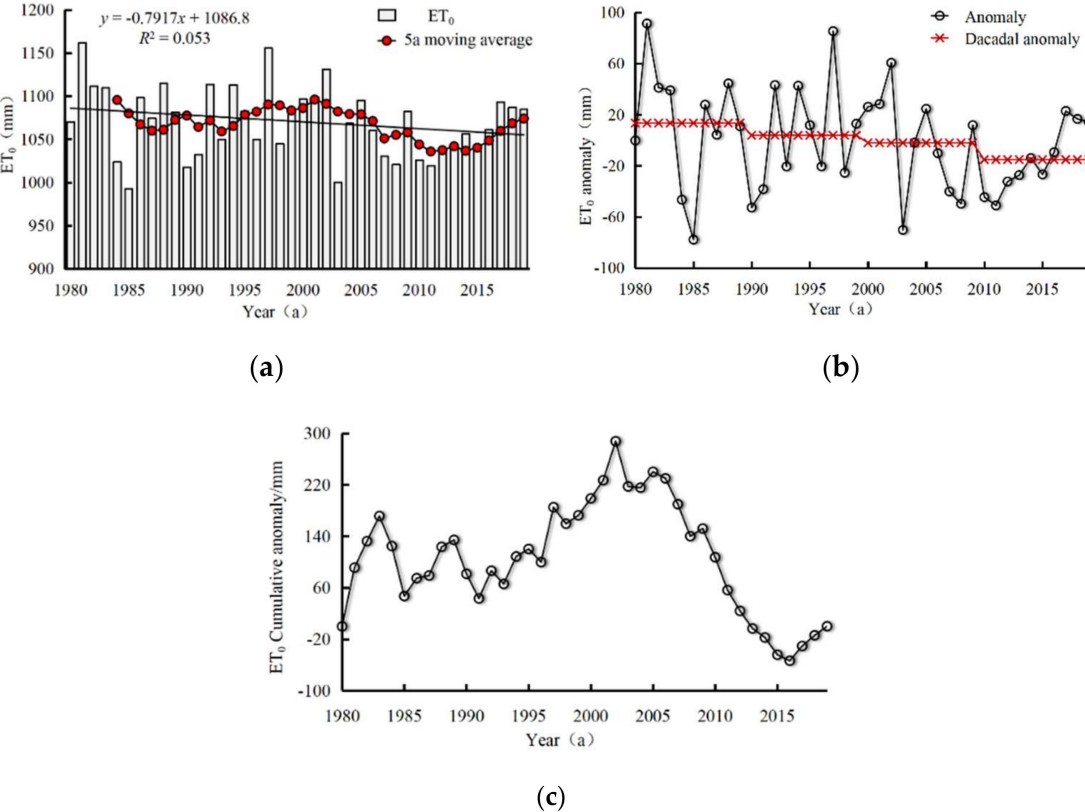

**Figure 4.** Anomaly and accumulated anomaly of annual evapotranspiration ($ET_0$), (**a**) 5a moving average of annual $ET_0$; (**b**) anomaly and decadal anomaly of annual $ET_0$; (**c**) accumulated anomaly of annual $ET_0$.

The seasonal difference was also analyzed. Table 2 show that the spring $ET_0$ had a slow upward trend, while the other three seasons showed a downward trend. The highest average annual $ET_0$ in summer was 406.51 mm; the lowest in winter was 107.40 mm. The average spring and autumn $ET_0$ were 334.32 and 222.24 mm, respectively. Overall, summer and spring $ET_0$ was greater than autumn and winter $ET_0$.

**Table 2.** Average and climatic inclinations of annual and seasonal evapotranspiration ($ET_0$).

| Time | Maximum Value/mm | Minimum Value/mm | Average Annual/mm | Climate Tendency Rate mm/10 a |
|---|---|---|---|---|
| Spring | 377.95 | 291.16 | 334.32 | 2.26 |
| Summer | 470.46 | 364.01 | 406.51 | −3.16 * |
| Autumn | 250.01 | 195.93 | 222.24 | −5.33 * |
| Winter | 125.02 | 83.32 | 107.40 | −1.57 |
| Annual | 1161.85 | 992.77 | 1070.55 | −7.92 * |

Note * indicates a significant level of $\alpha = 0.05$.

### 3.4. Spatial Distribution of $ET_0$

The $ET_0$ of 33 sites (86%) in Shandong Province showed a downward trend (Figure 5a). Zibo, Laizhou, and Qingdao's $ET_0$ showed a large negative climate tendency rate of −39.18, −34.49, and −25.87 mm/10 a, respectively. Five sites showed an upward trend (14%), among which the $ET_0$ of Heze, Wendeng, and Jiaozhou showed a large upward trend, of 30.07, 23.09, and 19.43 mm/10 a, respectively. Overall, the spatial distribution of $ET_0$ from 1980 to 2019 was not uniform. As shown in Figure 5b, the high-value interannual $ET_0$ areas were mainly distributed in northern Shandong, central Shandong, and the Jiaodong areas. The values in the south were relatively small. The spatial distribution characteristics of $ET_0$ at each local site varied because of their geographical location and climate environment. The high-value areas were mainly distributed in Jinan, Zhangqiu, and Weihai, with $ET_0$ values of 1257.35, 1185.34, and 1157.99 mm. Jinan and Zhangqiu are located on Mount Tai, with longer *SSD* and lower rainfall, so the $ET_0$ was higher, while Weihai is located on the shore of Huang Hai, which is affected by ocean circulation and had high *WS* and humidity, so the $ET_0$ was higher. The areas with low $ET_0$ values were mainly distributed in Chengshantou and Taishan, with values of 949.42 and 958.45 mm. Chengshantou is located on the eastern end of Jiaodong Peninsula, facing the sea, and the low *T* lead to low $ET_0$, while Taishan had low $ET_0$ because of the terrain and abundant rainfall on the Taishan mountain.

The spatial distribution of seasonal $ET_0$ was the highest in summer, with a variation range of 307.0−461.8 mm. With the exception of three sites (8%), which showed an upward trend, the remaining 35 sites (92%) had a downward trend. The highest value of summer $ET_0$ appeared in Jinan and Zhangqiu, at 461.9 and 448.4 mm respectively. The lowest value appeared in Chengshantou, with an average of 306.7 mm. The $ET_0$ changes in spring and autumn were 247.9−414.5 mm and 195.5−271.1 mm, respectively. About half the stations had a decreasing trend in spring, among which Zhangqiu and Heze had a large climate tendency rate of 13.8 and 13.1 mm/10 a, while Laizhou and Hanting had a small climate tendency rate of −9.0 and −9.8 mm/10 a, respectively. The spatial distribution of high and low $ET_0$ values in spring and summer was approximately the same. The number of stations with a rising trend in autumn $ET_0$ was one higher than that in summer. Wendeng and Jiaozhou had higher climate tendency rates of 8.1 and 5.9 mm/10 a, while Zibo and Hanting had smaller climate tendency rates of −12.7 and −13.2 mm/10 a, respectively. The highest autumn $ET_0$ value appeared in Chengshantou, Long Island, and other places at 271.1 and 270.2 mm, respectively. The lowest value appeared in Feicheng, with an average of 195.4 mm. The winter $ET_0$ was the lowest, with a range of 90.05−181.4 mm. Fifteen sites (39%) showed an upward trend, and the remaining 23 sites (61%) showed a downward trend. The highest value of winter $ET_0$ appeared in Longkou (181.5 mm), while the lowest value appeared in Jiyang, with an average of 90.4 mm. On the whole, the $ET_0$ of spring and summer decreased gradually from north to south, and the $ET_0$ of autumn and winter increased gradually from west to east.

Figure 6 shows that, overall, the average proportion of spring, summer, autumn, and winter $ET_0$ over a whole year was 29%, 40%, 21%, and 10%, respectively. The contribution of summer $ET_0$ was the highest, and the decrease in summer $ET_0$ was the main reason for the decrease of annual average $ET_0$. The largest seasonal $ET_0$ difference was at Jinan station, reaching 349.9 mm. The $ET_0$ difference between summer and winter was only 157.8 mm, and the seasonal $ET_0$ change was not obvious. The seasonal $ET_0$ in the hilly area in southwest Shandong was quite similar, but the seasonal $ET_0$ was very different in the Jiaodong area.

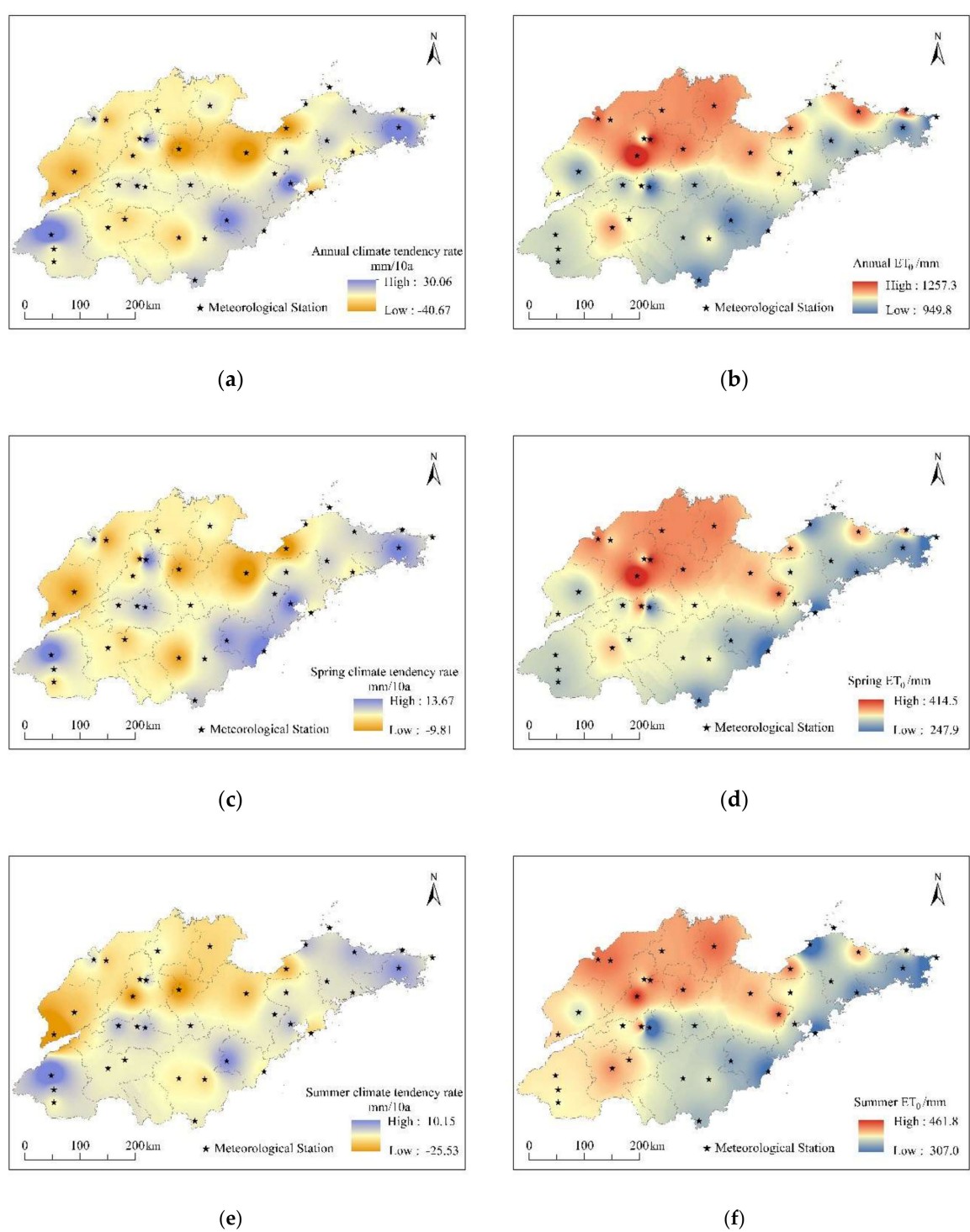

(**a**)

(**b**)

(**c**)

(**d**)

(**e**)

(**f**)

**Figure 5.** *Cont.*

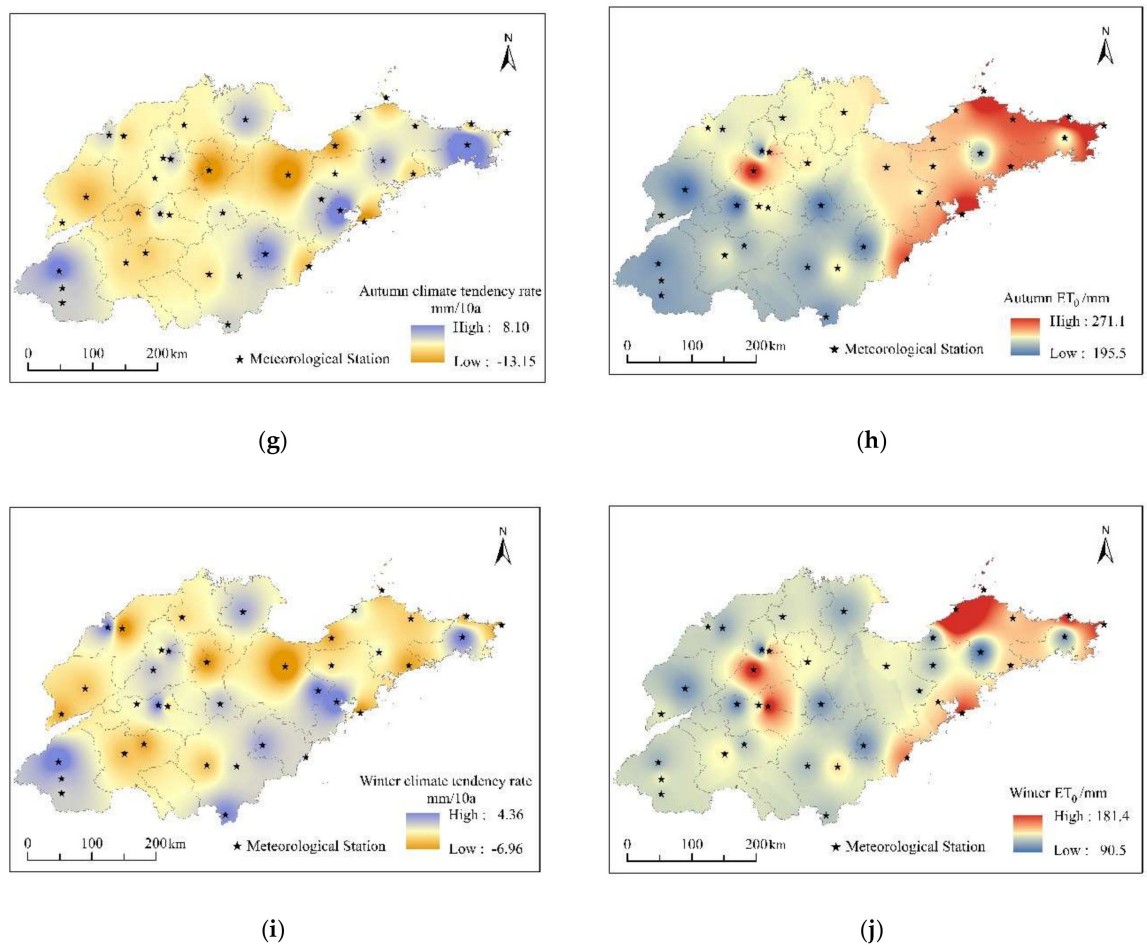

**Figure 5.** Spatial distribution of annual and seasonal evapotranspiration ($ET_0$) (**b**) annual, (**d**) spring, (**f**) summer, (**h**) autumn, and (**j**) winter. Climate tendency rate of annual and seasonal evapotranspiration ($ET_0$) (**a**) annual, (**c**) spring, (**e**) summer, (**g**) autumn, and (**i**) winter.

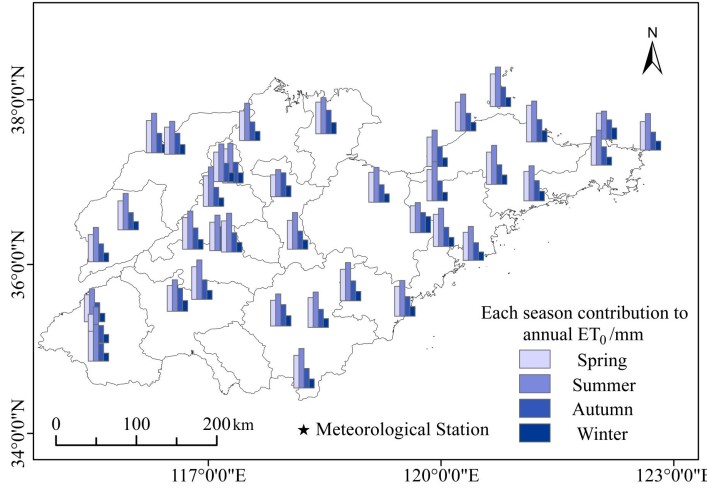

**Figure 6.** Spatial distribution of the contribution of seasonal evapotranspiration to annual $ET_0$.

### 3.5. Mann–Kendall Trend Test

To explore the reasons for the overall downward trend of annual $ET_0$ in Shandong Province, the M–K trend method was used to test the annual $ET_0$. The UF curve represents the time series,

and the UB curve represents the inverse time series (Figure 7). If the value of the UF curve is greater than 0, the time series shows an upward trend. The UF and UB intersection point with the line showing significance at 0.05 indicates an effective change. Figure 7 shows that the UF and UB curve of the interannual $ET_0$ of the study area intersected in 2002, indicating that the $ET_0$ change occurred in that year. The calculated decrease was 130.74 mm.

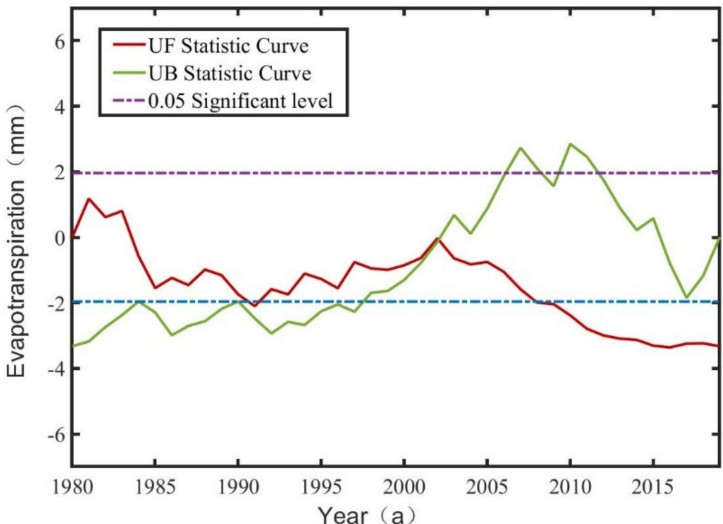

**Figure 7.** The result of Mann–Kendall (M–K) trend test.

To study whether the declining trend in $ET_0$ was related to meteorological factors, a trend test for $ET_0$ before, after, and during the change in relation to each meteorological factor was carried out, (Table 3). Prior to the change (1980−2002), $ET_0$ and other meteorological factors did not pass the test of significance at $\alpha = 0.05$, except for *SSD*. The *T* and *RH* showed an upward trend, and the other meteorological factors showed a downward trend. With the exception of *T*, the $ET_0$ and other meteorological factors showed a downward trend after the change (2002−2019) and over the whole period (1980−2019) and, except for *SSD*, the remaining factors were all statistically significant. The regional $ET_0$ showed a downward trend in the three periods, and the post-change statistic was −1.28, which was four times the level before the change. This indicated that the overall downward trend of $ET_0$ was largest in the period of 2002−2019. *T* showed an upward trend in all three periods, which was consistent with the *T* trend line in Figure 2a. *RH* was 0.27 > 0 between 1980 and 2002, indicating an upward trend, but between 2002 and 2019, the statistics were −1.44 < 0, indicating a downward trend. Overall, the average *RH* showed a downward trend. *WS* and *SSD* also decreased in the three periods.

**Table 3.** Trend test of evapotranspiration ($ET_0$) and meteorological factors in Shandong Province.

| Time | $ET_0$ | *T* | *RH* | *WS* | *SSD* |
|---|---|---|---|---|---|
| Before change (1980−2002) | −0.31 | 0.66 | 0.27 | −0.10 | −0.26 * |
| After change (2002−2019) | −1.28 * | 0.50 * | −1.44 * | −0.33 * | −0.07 |
| Whole period (1980−2019) | −1.35 * | 0.44 * | −0.68 * | −0.194 * | −0.25 * |

Note * indicates a significant level test of 0.05. *T*, *RH*, *WS*, and *SSD* represent daily mean temperature, average relative humidity, 2 m high wind speed, and sunshine hours, respectively.

## 4. Discussion

The fifth Intergovernmental Panel on Climate Change (IPCC) assessment report indicated that between 1880 and 2012, average global warming was 0.85 ± 0.2 °C. In response, research into regional climate systems in the context of global warming has become a focus [29]. The atmospheric



evapotranspiration capacity can be characterized by $ET_0$, which is a basic parameter for estimating crop water demand, and is an important index for evaluating the degree of regional drought, water consumption by vegetation, and water supply and demand balance [30]. As an important agricultural area in China, the spatiotemporal variation characteristics of $ET_0$ and meteorological factors in Shandong Province were analyzed in order to guide irrigation in the region, to enhance the adaptability of agricultural production to climate change, and to improve resilience to climate disasters.

With regard to the meteorological factors influencing $ET_0$ in Shandong, the average annual values of *T*, *RH*, *WS*, and *SSD* were 13.2 °C, 66.5%, 2.16 m/s, and 6.6 h respectively over the last 40 a. The overall trend of *RH*, *WS*, and *SSD* decreased with time, and *T* rose, indicating that the climate of Shandong Province has become warmer and drier. According to the background of global climate warming, this conclusion is consistent with that of Ma [31] on irrigation water demand and its influencing factors in Shandong Province. Dong et al. [32] found that the annual average $ET_0$ of the North China Plain was 1071.37 mm, and the $ET_0$ decreased over the past 53 years (−12.8 mm/10 a). This study found that, in the Shandong Province over the past 40 years, $ET_0$ overall showed a significant downward trend. The annual average was 1070.55 mm, and the decline rate was −7.92 mm/10 a. With the exception of spring $ET_0$, which showed an upward trend of 2.26 mm/10 a, the $ET_0$ of other seasons showed a downward trend, and the climatic tendency rates for summer, autumn, and winter were −3.16, −5.33, and −1.57 mm/10 a, respectively. The decrease in summer $ET_0$ was the main factor influencing the decrease of annual average $ET_0$, because of the high summer *T*. The $ET_0$ ratio of the summer $ET_0$ to that of the whole year was 40%, and the relative change rate was 26.2%, which was consistent with the seasonal analysis of $ET_0$ in Shandong Province by Dong et al. [33]. The M–K test showed that the trend in $ET_0$ changed in 2002 and the calculated decrease was 130.74 mm. On the one hand, the meteorological factors before and after the change point showed that *RH* had a slow upward trend at a rate of 0.27 before the change. On the other hand, *WS* showed a downward trend before and after the change point, and the downward trend was significant after 2002. The statistical values for *WS* were −0.10 and −0.33, respectively. The combined effect of these two factors—*RH* and *WS*—contributed to the downward trend in $ET_0$.

When studying the spatial distribution of $ET_0$ and its influencing factors in Shandong in the past 40 a, it was found that the spatial distribution of *RH* had obvious zonal characteristics, while the difference among *T*, *WS*, and *SSD* was related more to longitude. There were clear differences in the spatial distribution of meteorological factors. As is vividly shown in Figure 8, *T* has a downward trend in the X axis in both east and west directions, because *T* in the Jiaodong Peninsula is generally smaller than that of southwest Shandong. This is because the Jiaodong Peninsula is affected by the ocean monsoon climate because it is near the Yellow Sea and the Bo Sea, which reduces the average *T*. The *T* showed a similar U-shaped trend to the north and south of the Y axis, which is consistent with the lower *T* of Taishan and other places in the central Shandong area. This was due to the high terrain of Mount Tai, which resulted in an average *T* of only 14.3 °C. The *RH* along the meridian showed a U-shaped trend of high and low between the two sides. The Jiaodong area had a higher *RH* than the southwest Shandong mountain area because of its coastal location. *RH* showed a downward trend along latitudinal lines, with the largest *RH* in southern Shandong. As a whole, the *WS* showed an increasing trend from west to east. Except for the high *WS* caused by the terrain in Mount Tai, the other high-value centers appeared in the coastal Jiaodong Peninsula. The Jiaodong Peninsula had large *WS* because of the monsoon climate. The *SSD* showed a gradual increasing trend from southwest to northeast of Shandong, with a variation range of 5.9–7.3 h. The overall change was small, because the average sea surface of the Yellow Sea in the Qingdao area is above the National Vertical Datum. Except for the Taishan area, the overall elevation gap in Shandong was small, resulting in no clear change in *SSD*.

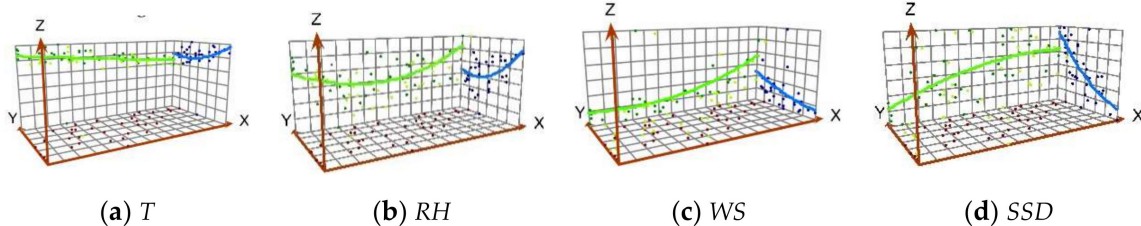

**(a)** *T*          **(b)** *RH*          **(c)** *WS*          **(d)** *SSD*

**Figure 8.** Spatial trends in meteorological factors. (**a**) daily mean temperature (*T*); (**b**) average relative humidity (*RH*); (**c**) wind speed at 2 m (*WS*); and (**d**) sunshine hours (*SSD*). Green curve is the fitting curve of the point projected from the meteorological data value to the Z,X plane, which represents the change trend in the longitude direction; similarly, the blue curve represents the change trend in the latitude direction.

On the whole, the distribution of spring and summer $ET_0$ in Shandong Province decreased from west to east and from south to north; the distribution pattern of autumn $ET_0$ was reversed for winter $ET_0$ (Figure 9). The spatial trend of interannual and seasonal $ET_0$ was clearly different. We analyzed the distribution of interannual and seasonal $ET_0$ further with the spatial trend map. The interannual $ET_0$ showed an approximate ∩-shaped pattern of low and middle values along the meridional direction, which followed the decreasing zonal trend from north to south. Therefore, the annual high-value area was mainly distributed in the middle and north of Lu, while the values in the south were relatively small. The spatial trend in $ET_0$ for spring and summer was similar to that of interannual $ET_0$, both of which decreased from west to east and from south to north. This was probably because the sum of $ET_0$ in spring and summer accounted for about 70% of the interannual $ET_0$ ratio. The change in $ET_0$ space in autumn and winter showed an obvious increasing trend from southwest to northeast. The autumn and winter periods were vulnerable to strong cold air, so *T* decreased and *WS* increased, which led to a trend of spatial change similar to that of *T* and *WS*., which was consistent with the quantification of causes for $ET_0$ in Shandong Province by Dong et al. [34].

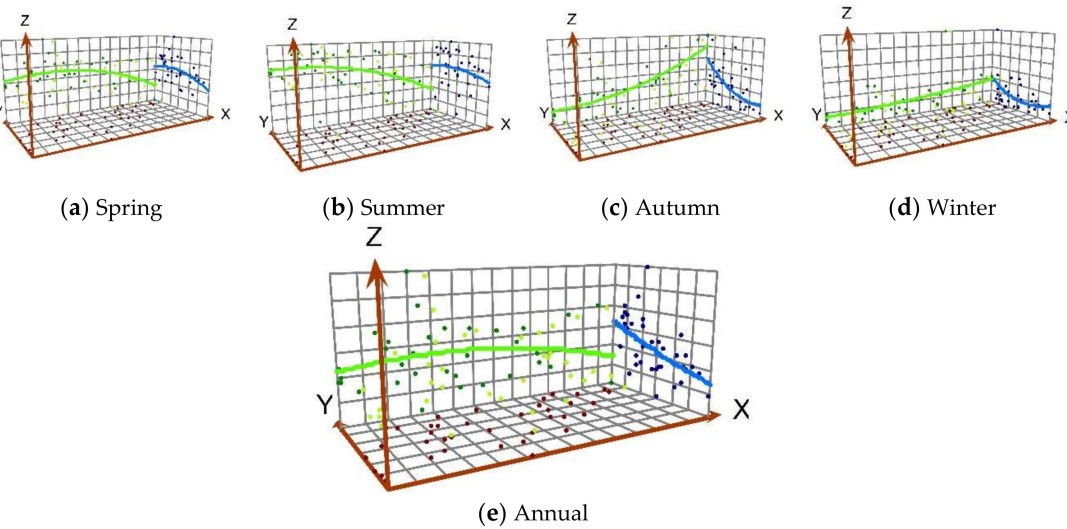

**(a)** Spring          **(b)** Summer          **(c)** Autumn          **(d)** Winter

**(e)** Annual

**Figure 9.** Spatial trends in annual and seasonal evapotranspiration ($ET_0$) (**a**) spring; (**b**) summer; (**c**) autumn; (**d**) winter; and (**e**) annual.

To explore the relationship between $ET_0$ and its influencing factors further, a Taylor correlation chart was drawn using $ET_0$ as reference point (Figure 10). With the exception of *RH*, the Pearson correlation coefficient between the other meteorological factors and the $ET_0$ was positive, that is, an increase in the values of the meteorological factors will lead to an increase in $ET_0$. The correlation between *T* and $ET_0$ was the most significant, reaching 0.93, and the correlation coefficients between

$ET_0$ and *WS* and *SSD* were 0.48 and 0.60, respectively. The *RH* was negatively correlated with $ET_0$, and the correlation coefficient was −0.49. This was consistent with the correlation analysis of $ET_0$ and meteorological factors in Shandong Province by Wang et al. [35] By comparing the Pearson correlation coefficients, it was found that *T* and *SSD* were the main meteorological factors affecting $ET_0$ change. As the global warming increases, the impact on the $ET_0$ of Shandong will gradually increase. At a next step, therefore, it is necessary to select the main grain crops, such as corn and wheat, and clarify the effect of climate change on the $ET_0$ of different growth periods in combination with the whole growth period. This will provide a theoretical basis for the construction of climate-adaptive cultivation models and irrigation methods for typical crops.

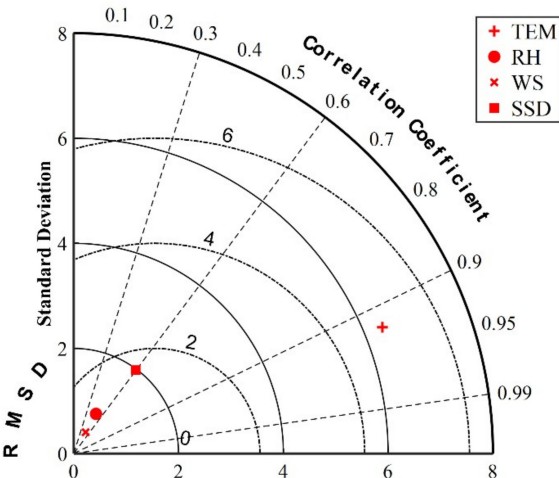

**Figure 10.** Taylor chart of correlation between evapotranspiration ($ET_0$) and meteorological factors. Daily mean temperature (*T*); average relative humidity (*RH*); wind speed at 2 m (*WS*); and sunshine hours (*SSD*).

## 5. Conclusions

(1) The average annual values for *T*, *RH*, *WS*, and *SSD* in Shandong Province from 1980 to 2019 were 13.2 °C, 66.5%, 2.16 m/s, and 6.6 h, respectively. The spatial distribution of *RH* had latitudinal zonal characteristics, and *T*, *WS* and *SSD* showed longitudinal variations.

(2) The average annual $ET_0$ of the last 40 years showed a downward trend, and the decline rate was −7.92 mm/10 a. From the 1980s to the 2010s, the decadal anomaly values decreased by 9.38, 5.88, and 13.11 mm per decade. Spring $ET_0$ showed a slow upward trend, the $ET_0$ of the other three seasons showed a downward trend. The spring, summer, autumn, and winter $ET_0$ as a proportion of that of the whole year was about 29%, 40%, 21%, and 10%. The highest contribution was from summer $ET_0$, and the decrease in summer $ET_0$ was the main reason for the decrease in annual $ET_0$. With regard to spatial distribution, spring and summer $ET_0$ generally showed a latitudinal distribution, decreasing from north to south, and autumn and winter $ET_0$ generally increased along a longitudinal gradient from west to east.

(3) The M–K trend test showed that interannual $ET_0$ change point in Shandong Province occurred in 2002, the reduction was about 130.74 mm. Annual $ET_0$ was positively correlated with *T*, *WS*, *SSD*, and negatively correlated with *RH*. *T* and *SSD* had the most significant correlation with $ET_0$, reaching 0.93 and 0.60, respectively.

From the point of view of spatiotemporal distribution, this paper analyzed the trend and correlation of $ET_0$ with other meteorological factors in Shandong Province over the past 40 years. The findings can help to estimate and guide irrigation water demand, increase agricultural production, and resist meteorological disasters. In the case of global climate change anomalies, quantitative analysis is needed in order to find out the leading meteorological factors affecting $ET_0$ change in Shandong

Province. This could provide a scientific basis for the regulation and control of agricultural production in Shandong Province.

**Author Contributions:** L.Z. collected the meteorological data; F.Z. and Z.L. analyzed the data; F.Z. and Z.L. wrote the paper; Z.L., L.Z., and J.Y. drew the figures for this paper; F.Z., Z.L., J.Y., K.S., and L.Y. reviewed and edited the paper. All authors have read and agreed to the published version of the manuscript.

**Funding:** This research was funded by the Major State Research Develop Program of China, grant numbers 2017YFD0301004 and 2016YFD0300201-3 and the Modern Agro-industry Technology Research System of Maize, grant number CARS-02-87.

**Acknowledgments:** We thank the Chinese meteorological data sharing service (http://data.cma.cn) for providing the meteorological data.

**Conflicts of Interest:** The authors declare no conflict of interest.

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
