# Peer review of "Spatiotemporal Distribution Characteristics of Reference Evapotranspiration in Shandong Province from 1980 to 2019"

_water, doi:10.3390/w12123495_

Round 1

Reviewer 1 Report

Fig. 2. It is not known whether the equations in the graphs are statistically significant.

Fig 4a. The downward trend is statistically insignificant. 5a moving average not described in the text. No significant downward trend until 2010. The value of the decade anomaly was: 1980s increase by 30 mm, 1990s increase by 6 mm, 2000s decrease by 9 mm, 2010s decrease by 13 mm. ET decreased until 2016 (Fig. 4c).

Seasonal variability of ET is unnecessary. Delete lines 211 to 219 and 269 to 277.

There are too few comparisons to the research of other authors in the discussion. Modify conclusion 2 according to the above comments.

Author Response

Dear Reviewer1:

Thank you for your comments on our manuscript entitled "Spatio-temporal distribution characteristics of reference evapotranspiration in Shandong province from 1980 to 2019" (WaterID:1021809). Those comments are very helpful for revising and improving our paper, as well as the important guiding significance to other research. We have studied the comments carefully and made corrections which we hope meet with approval. The main corrections are in the manuscript and the responds to the reviewers’ comments are as an attachment “Response to Reviewer1 Water.doc”(the replies are highlighted in blue ).

Once again, thank you very much for your constructive comments and suggestions which would help us both in English and in depth to improve the quality of the paper.

Kind regards,

Mr.Liu

Corresponding author : Ms.Zhangzhong

E-mail address: [email protected]

Reviewer 2 Report

If you want the reference crop evapotranspiration you need to calculate it multipliyng by kc crop coefficient (there are many studies about vitis vinifera i advice you to read and cite).

Line 36 - the symbol of reference crop evapotranspiration is ET0. Furthermore atmospheric evapotranspiration doesn't exist, write potential evapotranspiration or reference evapotranspiration. 

Why the study period is 1980-2019, you are out of the WMO prescriptions, justify why.

Why don't you give an overview of the most common interpolation methods  applied to climate ? Why do you use the idw?

Read and cite this papers:

Radial basis function:

Ahmed, K., Shahid, S., & Harun, S. B. (2014). Spatial interpolation of climatic variables in a predominantly arid region with complex topography. Environment Systems and Decisions34(4), 555-563.

Kriging:

Gentilucci, M., Materazzi, M., Pambianchi, G. et al. Temperature variations in Central Italy (Marche region) and effects on wine grape production. Theor Appl Climatol 140, 303–312 (2020).

EBK:

Zhang, Z., & Du, Q. (2019). A Bayesian Kriging Regression Method to Estimate Air Temperature Using Remote Sensing Data. Remote Sensing11(7), 767.

Introduction does not go into the state of the art properly.

line 300 - There are methods to assess climate change report it to a given year. Homogenization method e.g. Alexandersson's test.

Arcgis' trend analysis is used to discover trends along certain directions and possibly detract from the historical series by working on the residues. It is not the appropriate tool to certify the presence of trends, for that there are tests, such as the one you have used the Mann-Kendall.

Do not even report the angle where you saved the trend analysis on the GIS software.

There are some inaccuracies, you need to study the methods analysed more carefully.

What is the innovativeness of this study?

Author Response

Dear Reviewer2:

Thank you for your comments on our manuscript entitled "Spatio-temporal distribution characteristics of reference evapotranspiration in Shandong province from 1980 to 2019" (WaterID:1021809). Those comments are very helpful for revising and improving our paper, as well as the important guiding significance to other research. We have studied the comments carefully and made corrections which we hope meet with approval. The main corrections are in the manuscript and the responds to the reviewers’ comments are as an attachment “Response to Reviewer2 Water.doc”(the replies are highlighted in blue ).

Once again, thank you very much for your constructive comments and suggestions which would help us both in English and in depth to improve the quality of the paper.

Kind regards,

Mr.Liu

Corresponding author : Ms.Zhangzhong

E-mail address: [email protected]

Reviewer 3 Report

In Abstract, pag. 1, rows 19-20: in the sentence “… the spatio-temporal variations and trends in Shandong Province from 1980 to 2019.”, please include the location of the study area, e.g. “in Shandong Province, located on the east coast of China”.

In Abstract, pag. 1, rows 20: in the sentence “(1) The ET0 from 1980 to 2019 was…” please include “average annual ET0”.

In Abstract, pag. 1, row24: please defined the ET0 ratio.

In chp. 1. Introduction, pag. 2, row 51: please add the location of the Heilongjiang Province in China.

In chp. 1. Introduction, pag. 2, row 53: please add the location of the Jiangsu Province in China.

In chp. 1. Introduction, pag. 2, row 56: please add the location of the hilly region of central Sichuan in China.

In chp. 1. Introduction, pag. 2, row 62: please add the location of the Loess Plateau of China in China.

In chp. 3. Results, subchp. 3.5. Mann-Kendall trend test, pag. 10, rows 280-281”: in the sentence „The UF curve represents the time series, and the UB curve represents the inverse time series (Figure 8).”, please define more clearly the UB curve.

Author Response

Dear Reviewer3:

Thank you for your comments on our manuscript entitled "Spatio-temporal distribution characteristics of reference evapotranspiration in Shandong province from 1980 to 2019" (WaterID:1021809). Those comments are very helpful for revising and improving our paper, as well as the important guiding significance to other research. We have studied the comments carefully and made corrections which we hope meet with approval. The main corrections are in the manuscript and the responds to the reviewers’ comments are as an attachmentResponse to Reviewer3 Water.doc”(the replies are highlighted in blue ).

Once again, thank you very much for your constructive comments and suggestions which would help us both in English and in depth to improve the quality of the paper.

Kind regards,

Mr.Liu

Corresponding author : Ms.Zhangzhong

E-mail address: [email protected]

Round 2

Reviewer 2 Report

The answers you provided are not exhaustive. Please write more in the text

Author Response

Dear Reviewer2:

Thank you for your comments on our manuscript entitled "Spatio-temporal distribution characteristics of reference evapotranspiration in Shandong province from 1980 to 2019" (WaterID:1021809). Those comments are very helpful for revising and improving our paper, as well as the important guiding significance to other research. We have studied the comments carefully and made corrections which we hope meet with approval. The main corrections are in the manuscript and the responds to the reviewers’ comments are as an attachment “Response to Reviewer2 Water Round 2.doc”(the replies are highlighted in blue ).

Once again, thank you very much for your constructive comments and suggestions which would help us both in English and in depth to improve the quality of the paper.

Kind regards,

Mr.Liu

Corresponding author : Ms.Zhangzhong

E-mail address: [email protected]
